# Influence of geomagnetic activity on mesopause temperature over Yakutia

Galina Gavrilyeva, Petr Ammosov

Yu. G. Shafer Institute for Cosmophysical Research and Aeronomy SB RAS, 677098, Yakutsk, Russian Federation

*Correspondence to*: gagavrilyeva@ikfia.ysn.ru

**Abstract.** The long-term temperature changes of the mesopause region at the hydroxyl molecule OH (6-2) nighttime height and its connection with the geomagnetic activity during the 23rd and beginning of the 24th solar cycles are presented. Measurements were conducted with an infrared digital spectrograph at the Maimaga station (63°N, 129.5°E). The hydroxyl rotational temperature (TOH) is assumed to be equal to the neutral atmosphere temperature at altitude of ~87 km. The average

temperatures obtained for the period 1999 to 2015 are considered. The season of observations starts at the beginning of August and lasts until the middle of May. The maximum of the seasonally averaged temperatures is delayed by 2 years relative to the maximum of ~~flux of~~solar radio emission ~~from the Sun with a~~ flux (wavelength of 10.7 cm~~,~~), and correlates with a change in geomagnetic activity~~.~~ (Ap-index ~~as a measure of geomagnetic activity is taken.~~). Temperature grouping in accordance with the geomagnetic activity level showed that in years with high activity (Ap> 8), the mesopause temperature from October to

February is about ·10 K higher than in years with low activity (Ap <= 8). Cross-correlation analysis showed no temporal shift between geomagnetic activity and temperature. The correlation coefficient is equal 0.51 ~~±at the~~ 0.~~1 at 95% confidence~~05 level of significance.

## Introduction

Long-term changes in the state of the mesopause, such as the linear trend and the fluctuations associated with the 11-year cycle

of changes in solar activity, are investigated by different methods. In the review Beig et al. (2008) ~~show that, according to~~lists numerous studies~~,~~ showing that the response of the mesosphere/low thermosphere temperature to the change in solar activity reaches 4-5 K / 100SFU, where SFU is the solar radio flux at a wavelength of 10.7 cm in $10^{-22}$ W M$^{-2}$ Hz$^{-1}$~~.~~ (F10.7). Tang et al. (2016) estimated the change in the temperature of the mesopause from 2002 to 2015 using the measurements of the SABER radiometer ~~installed on~~onboard the TIMED satellite. They showed that the average global response is about 5 K / 100 SFU,

~~which does not contradict~~in agreement with the results given in the review of Beig et al. (2008). The response of the temperature to the change in the flux of radio emission F10.7 at high latitudes is greater than at the middle latitudes and reaches up to 7-10 K / 100SFU.

Previously, according to data obtained from 1997 to 2000 at the Maimaga station, we found the temperature response equal to 11 K / 100SFU (Gavrilyeva and Ammosov, 2002). This study only used a very short period of observations which coincided

with the maximum of solar activity. Further, Ammosov et al. (2014) presented the results of data analysis obtained in a time interval comparable to the solar cycle duration from 1999 to 2013. Analysis showed that the temperature change follows the solar activity change with 25 months delay. The temperature response at the delay of 25 months reaches 7 K/100 SFU.

It is known that the geomagnetic activity maximum lags behind the solar radiation maximum including the index F10.7. ~~As a measure of geomagnetic activity widely available index Ap was used. Both indexes were acquired from the National~~

~~Geophysical Data Center, NGDC (ftp://ftp.ngdc.noaa.gov/STP).~~ The radio emission flux F10.7 and Ap-index of magnetic disturbance changes over the last 4 cycles of solar activity is shown in Figure 1. As a measure of geomagnetic activity, the widely available index Ap was used. Both indices were acquired from the National Geophysical Data Center, NGDC (ftp://ftp.ngdc.noaa.gov/STP). As can be seen from the Figure 1~~,~~ Ap-index changes follow the F10.7 changes with a lag of 2-

years. ~~Therefore~~As this is similar in scale to the observed delay of 25 months, it was logical to assume that the long-term temperature fluctuation of the subauroral mesopause correlates with the change in geomagnetic activity.

Geomagnetic activity can change the composition, dynamics and thermal state of the upper latitude atmosphere through the of energetic particles precipitation (EPP). In the last decade, many papers have been published on the atmosphere response to the proton and electron fluxes with various energies. In these papers, two ways of the geomagnetic activity influence to the atmosphere temperature were discussed.

First way is direct effect on the temperature and dynamics. The geomagnetic storm is followed in the atmosphere by ionization, excitation, Joule heating and dissociation processes (Lastovicka, 1996, Burns et al., 2014, Xu et al., 2013). There are some evidences of the upper atmosphere temperature change during EPP. Xu et al., (2013) investigated the longitudinal temperature structure in the lower thermosphere using the SABER / TIMED and Envisat / MIPAS (Michelson Interferometer for Passive Atmospheric Sounding) data obtained from 2008 to 2009. Study of satellite measurements reveals that the maximum of the diurnally averaged temperature in the lower thermosphere is near the longitude of the magnetic pole in both the Northern and Southern Hemispheres. Authors suggested that this structure of the diurnally averaged temperature in the lower thermosphere is most likely related to auroral heating, which occurs in the auroral region near the magnetic poles. von Savigny et al., (2007) observed a significant decrease in the noctilucent clouds occurrence rate in the southern polar mesopause region made with SCIAMACHY (an imaging spectrometer installed on satellite Envisat), immediately after the onset of the enhanced solar particle precipitation on January 16, 2005. Simultaneously, the instrument Microwave Limb Sounder (MLS) on board of NASA's satellite AURA registered the atmosphere temperature increase at an altitude of 85 km. Hocke (2017) studied the temperature measurement with the MLS on AURA during the proton event on November 7-10, 2004. He found the temperature increasing of the polar mesosphere by 5-10 K while the polar stratosphere temperature decreased. Analyses of SABER / TIMED temperature data made by Chang et al., (2009), and Jiang et al., (2014) showed that periodic oscillations of the low thermosphere temperature had good correlations with oscillations in geomagnetic activity.

As well as the direct effect of EPP, there is also an indirect effect on the atmosphere from particle penetration whose energy is deposited mainly in the thermosphere and upper mesosphere. Studies showed that particle precipitations through a cascade of dissociation, ionization and recombination processes create odd nitrogen ($NO_x$) and odd hydrogen ($HO_x$) in the high latitude thermosphere and mesosphere. $HO_x$ is relatively short-lived (of the order of days) leading mostly to local effects, while $NO_x$ can lead to both short and long term (order of months) catalytic ozone destruction in the stratosphere. These effects may further couple to atmospheric dynamics and propagate downwards by changing polar winds and atmospheric wave propagation through wave—mean flow interaction (Krivolutsky et al., 2006, Baumgaertner et al., 2009, 2011, Semenuk et al., 2011, Arsenovic et al., 2016, Karami et al., 2015, Randall et al., 2007).

It is known that the rotational temperature of the hydroxyl corresponds to the temperature of the neutral atmosphere at the height of its radiation (~ 87 km). Consequently, the effect of geomagnetic activity on the temperature of the atmosphere can be investigated from the change in the rotational temperature of the hydroxyl. The purpose of this paper is to ~~investigate the connection between the~~find geomagnetic ~~activity change (Ap index) and the~~signatures in night measurements of OH rotational temperature ~~based on night measurements~~ obtained for the period August 1999 to May 2015. As a measure of geomagnetic activity, the widely available index Ap was used.

**Instrumentation and measurement technique**

Mesopause (80-100 km) is the atmosphere region where the mesosphere borders on a thermosphere ~~(80-100 km) and a temperature minimum is located.~~ The radiating layer of excited hydroxyl (OH) is located at about 87 km in mesopause region. The ~~activated~~exited hydroxyl molecule ~~commits~~experiences $2 \cdot 10^4$ s$^{-1}$ collisions before radiation, which is sufficient for thermalization with the surrounding medium. Therefore, the OH rotational temperature calculated from the night sky spectra indicate the neutral atmosphere temperature at the radiation height (Khomich et al., 2008).

The OH(6-2) rotational temperature data (TOH) for the presented paper were obtained with the infrared spectrograph (Ammosov and Gavrilyeva, 2000). The spectrograph was installed at the optical station of Maimaga (geographic coordinates are 63° N, 129.5° E) which is, geomagnetic coordinates are 58° N, 202° E ) located at a distance of about 120 km to the north of Yakutsk, Russia. Observations were carried out in cloudless and moonless nights, with the sun at least 9° below the horizon. For the analysisThe atomic oxygen line which arises at high auroral activity superimposes on OH(6-2) spectrum. To avoid systematic errors in evaluating the temperature because of this, the data obtained during moonless time and in the absence of aurora were selected for the analysis. The location of the observation station makes it possible to perform measurements only from the beginning of August to the middle of May since the summer mesopause is constantly sunlit at the Maimaga latitude. The method for estimating the rotational temperature of molecular emissions is based on the least squares fit of model spectra constructed with regard to the instrument function for different previously specified temperatures to an actually measured spectrum (Ammosov and Gavrilyeva, 2000). The temperature corresponding to that model spectrum whose deviation, which deviates least from the actual one is lessreal spectrum, by not more than the registration noise, is considered to correspond most closelyas a best fit to the reality; and the real hydroxyl rotational temperature determined based on this spectrum corresponds to the temperature at the mesopause height.. The estimates indicate that random errors in measuring the temperature measurements vary fromare typically 2 to 10 K, depending on the signal-to-noise ratio. As the estimations show the errors of temperature measuring are in the range of 2-5 K depending on the signal to noise ratio level. Since different published transition probabilities lead to temperature differences up to 12 K (Turnbull and Lowe, 1989; Greet et al., 1998) all the data are analyzed using the same Einstein coefficients by Mies (1974), for consistency.

**Results**

The rotational temperature data set comprises 2864 nightly average temperatures obtained from August 1999 to May 2015. The measurements of the nightglow spectrum are conducted from the beginning of August to the beginning of May. The longest night data series are registered in the winter. The number of measurements per month varies from 10 to 25 nights. The TOH and F10.7 index average values for the measurement season (from August to May) for 1999-2015 are plotted in Figure 2a. The TOH and Ap-index mean values variations are shown in Figure 2b. The average values of the F10.7 index and Ap-index were calculated in the days that coincided with the TOH measurements at the Maimaga station. As can be seen from the Figure 2, the TOH inter annual variation is delayed relative to the F10.7 change and is more consistent with the Ap-index variation. The correlation coefficient of TOH and Ap-index is equal 0.51 ± 0.1 at 95% confidence level0.51. The significance of correlation coefficient was tested with 14 degrees of freedom T-test. The critical value of correlation coefficient is 0.46 at the 0.05 level of significance. TOH is not significantly correlate with F10.7, because correlation coefficient 0.36 is less than critical value. The correlation coefficient increases to 0.65 when F10.7 leads the temperature by 2 years.

The night temperature means were divided into two groups for further analysis. FirstThe average AP in the observation interval of about 8 was chosen as the transition value. The first group includes the measurements which were conducted at the season with high geomagnetic activity when average Ap-index > 8. SecondThe second group consists of night TOH measured during the season with Ap-index <= 8. The number of observations per month in two groups is shown in Figure 3. The seasonal distribution of measurements is approximately the similar. A monthly mean TOH in geomagneticgeomagnetically active years (Ap> 8) and in geomagnetic quiet years (Ap <= 8) are plotted in Figure 4. The mesopauseThe results show higher monthly mean OH temperature with high Ap (>8) than with lower Ap (<=8) from October to Februarythrough January. The difference is higher in the geomagnetic active years in comparison with the geomagnetic quiet years.about 10 K (i.e. 10.5K±1.4K, or 9.6K±1.4K, if Feb is included). There is no dependence of the TOH on the level of geomagnetic activity in autumn and spring. However, it should be noted that at this period the number of observations is not large.

**Discussion**

~~In the last decade, many papers have been published which were devoted to the study of the atmosphere response to the proton and electron fluxes with various energies. Model calculations showed that particle precipitations through a cascade of dissociation, ionization and recombination processes create nitrogen (NOx) and hydrogen (HOx) oxides in the high latitude thermosphere and mesosphere. HOx is relatively short-lived (of the order of days) leading mostly to local effects, while NOx can lead to both short and long term (order of months) catalytic ozone destruction in the middle atmosphere (Krivolutsky et al., 2006, Baumgaertner et al., 2009, 2011). Observations from satellites show that energetic particles precipitation change the nitrogen oxides amount in the atmosphere. Randall et al. (2007) investigated the energetic particles precipitation effect on the southern hemisphere stratosphere from satellites measurements for the period 1992 to 2005. It was shown that the amount of NOx produced by the energetic particles precipitation corresponds to the level of geomagnetic activity. Baumgaertner et al. (2009, 2011) used the atmospheric chemistry general circulation model ECHAM5/MESSy to simulate polar air temperature effects of geomagnetic activity variations. The researchers calculated NOx fluxes formed by low-energy electrons precipitations using the average annual Ap from 1991 to 2005. Model average annual NOx concentrations were compared with the nitrogen oxides concentrations computed by Randall et al., (2007) from HALOE radiometer measurements on board the UARS satellite. The measured and model average annual concentrations of nitrogen oxides are almost identical (see Figure 1. in Baumgaertner et al., 2009). Thus, the authors have convincingly demonstrated that the electron precipitations can be the sources of nitrogen and hydrogen oxides in the upper and middle atmosphere. In the further paper of Baumgaertner et al. (2011) showed that, strong geomagnetic activity and the associated NOx enhancements lead to polar stratospheric ozone loss. Compared with the simulation with weak geomagnetic activity, the ozone loss causes a decrease in ozone radiative cooling and thus a temperature increase in the polar winter mesosphere. (Figure 9 in Baumgaertner et al., 2011). A similar effect of the ozone loss due to energetic particles precipitation on the atmosphere temperature and dynamics in other models has been obtained (Semenuk et al., 2011, Karami et al., 2015, Arsenovic et al., 2016,). There is a publication series (Lu et al., 2008, Sëppala et al., 2009, Seppala~~There are several publications (Lu et al., 2008, Seppälä et al., 2009, Seppälä et al., 2013), where the authors investigated the geomagnetic activity effect in the atmosphere based on the meteorological measurements ERA-40 and ERA interim data set. The authors studied the atmosphere climatology from 1000 hPa to 1 hPa separately in the years with high and low geomagnetic activity. They found that high geomagnetic activity can drive a strengthening of the Northern Hemisphere polar vortex, with warming in the polar upper stratosphere and cooling below. Meteorological data analysis shows that the upper stratosphere warming starts in beginning of December and lasts until March (~~Sëppala~~Seppälä et al., 2013). The heating descends downwards during winter. Karami et al. (2015) investigated the thermal and dynamic response of the middle atmosphere to ozone concentration losses due to ~~energetic particles precipitation~~EPP using the chemistry–climate general circulation model EMAC. The results of simulations show that as winter progresses the temperature anomaly ~~move~~moves downward with time from the mesosphere/upper stratosphere to the lower stratosphere. A similar downwards descending signal (in the same model) is already demonstrated by Baumgaertner et al (2011) using Geopotential height anomalies.

The temperature difference in the geomagnetic active years in comparison with the geomagnetic quiet years has been observed since October to February in our measurements. It should be noted that model and experimental researches of meteorological parameters are limited to a height of 80 km. The hydroxyl radiating layer is located in the mesopause region (~ 87 km). ~~Therefore, warming~~As indicated above, the temperature difference in the geomagnetic active years in comparison with the geomagnetic quiet years has been observed since October to February in our measurements ~~has to~~. It should be ~~detected~~noted that model and experimental researches of meteorological parameters are limited to a height below ~80 km. The hydroxyl radiating layer is located 7 km higher. Stratwarm effects are known to propagate downward, so that the OH temperature effect should be expected to occur earlier. ~~The energetic~~ than model results obtained for 80 km, and below. However, one cannot be sure that the observed temperature difference is the result of an indirect impact. The temperature signal can be related to auroral heating, or in situ ozone depletion caused short-time HO$_x$ enhancement. Unfortunately, we cannot investigate the direct effect

of precipitating particles precipitation change, since a line of atomic oxygen is superimposed on the hydroxyl spectrum in geomagnetic active days. Such spectra are excluded from the analysis.

The EPP changes temperature and dynamics in the winter polar atmosphere as shown in the above studies. MostAlso, most of the measurement of the mesopause region temperature at our latitude is also carried out in the winter. Figure 5 shows the F10.7 and Ap-index averages variations in January from 1975 to 2016. The regular measurements of the mesopause region temperature began approximately in these years. Unlike the previous solar cycles, it is clearly seen that F10.7 maximum leads Ap-index maximum by about 2-3 years in the 23rd solar cycle. It should be noted, that in our research the influence of the solar irradiance and the long-term linear trend on the mesopause temperature is not studied. In orderThe data of several solar cycles is necessary because to separate correctly the influence of these components, the data of several solar cycles is necessary.

~~The energetic particles precipitation can have a direct effect on the mesopause region temperature and dynamics. von Savigny et al., (2007) observed a severe decrease in the noctilucent clouds occurrence rate in the southern polar mesopause region made with SCIAMACHY on Envisat satellite immediately after the onset of the enhanced solar particle precipitation on January 16, 2005. At the same time, the atmosphere temperature increase at an altitude of 85 km are employed with the MLS on AURA. Temperature measurement with the MLS on AURA during the proton event on November 7-10, 2004 showed the polar mesosphere temperature increase by 5-10 K while the polar stratosphere temperature decreased (Hocke, 2017). However, other researches showed that there may be another indirect mechanism. Dynamical interaction between the mean flow and planetary waves in the stratosphere play an important role in transferring the geomagnetic activity induced effects (Arsenovic et al., 2016, Seppala et al., 2013, Karami et al., 2016).~~

## Conclusions

The data set of the hydroxyl emission airglow comprises 2864 nightly average temperature values obtained from August 1999 to May 2015 at the subauroral Maimaga station are considered. The measurements of rotational temperature of OH(6-2) at Maimaga subauroral station for the August 1999 to May 2015 period were studied in search for geomagnetic activity effect. Correlation between seasonally averaged TOH and geomagnetic activity index Ap is statistically significant and is equal to 0.51.

The winter polar mesopause is approximately 10K10 K warmer in the years with high geomagnetic activity (Ap>8), than in the years with low geomagnetic activity (Ap<=8). Warming of the mesopause starts in October and lasts until February, which is about 1-2 months earlier than the warming in the stratosphere. It can be explained by altitude difference between mesopause (87 km) and 1 hPa (50 km) level where the average onset of warming was noticed (Sëppalastratospheric warmings is observed (Seppälä et al., 2013). Warming signal moves down from high altitude to low one. Thus, measurements of the mesopause Nevertheless, it cannot be ruled out that the temperature revealedrise of the mesopause region heatingupper mesosphere in geomagneticgeomagnetically active periods foryears is due to the first timein situ effect of EPP.

## Acknowledgments.

The Russian Foundation for Basic Research supported the reported study was funded by RFBR according to the research projects No. 17-05-00855 A, 15-05-05320 A.

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

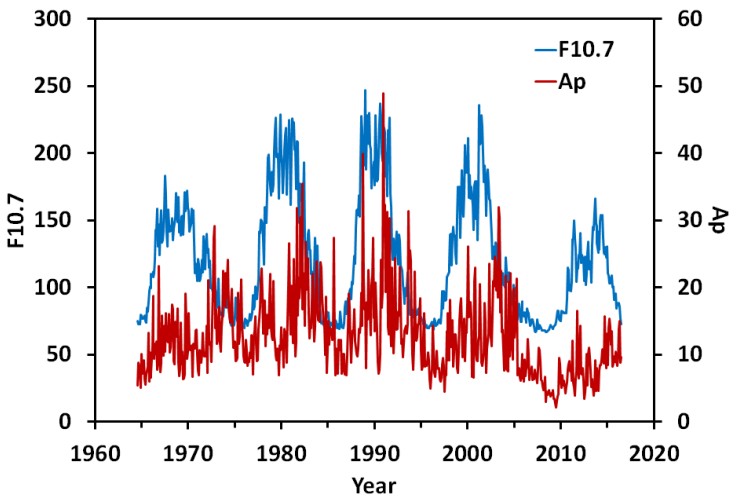

Figure1: Monthly mean F10.7 and Ap for 1965-2016. Both indexesindices were acquired from the National Geophysical Data Center, NGDC (ftp://ftp.ngdc.noaa.gov/STP).

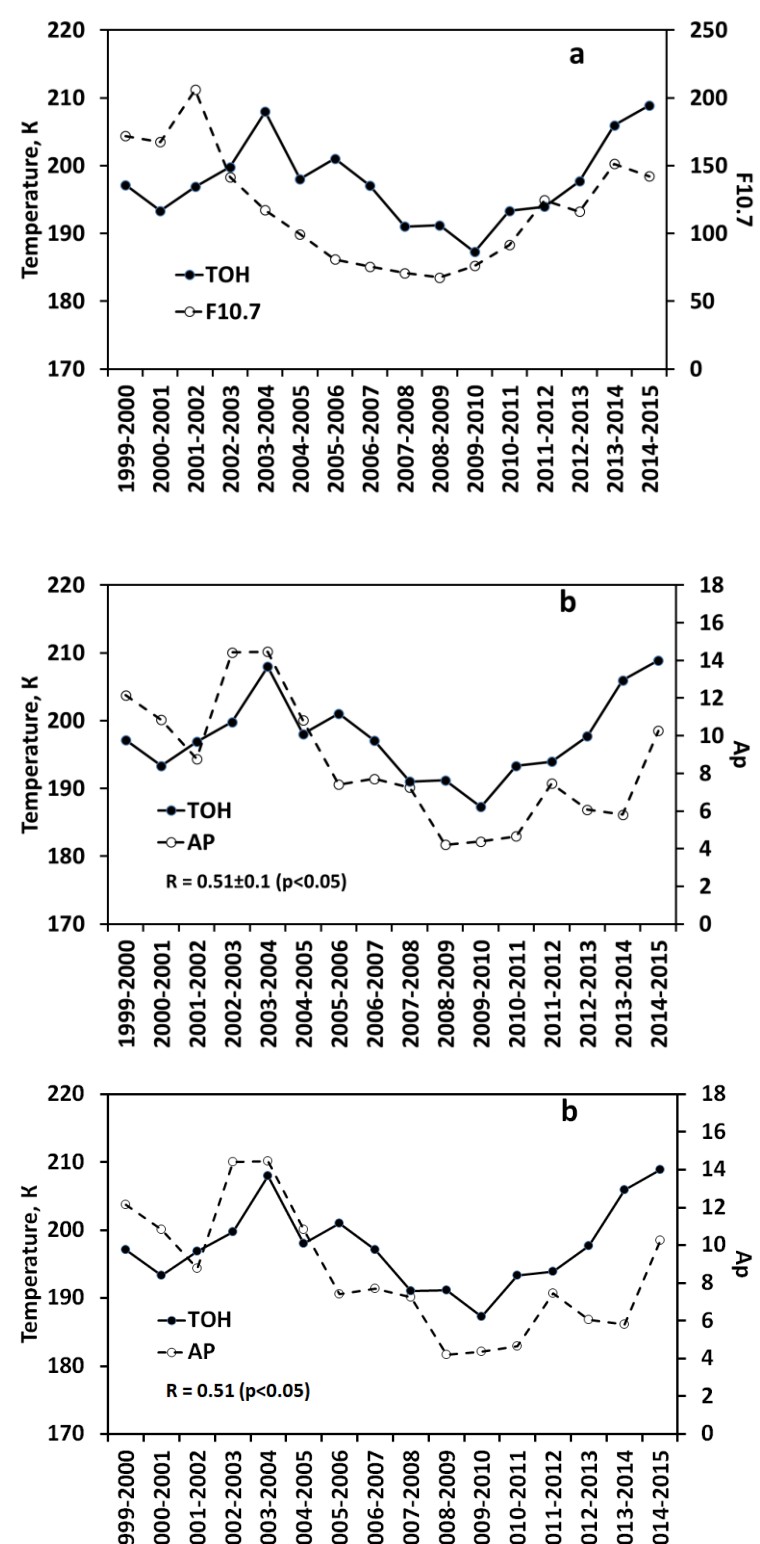

5    Figure 2: (a) Seasonally averaged TOH and F10.7 index (from August to May) for 1999-2015. (b) The TOH and Ap-index mean values for 1999-2015. The average values of the F10.7 index and Ap-index were calculated in the days that coincided with the TOH measurements at the Maimaga station.

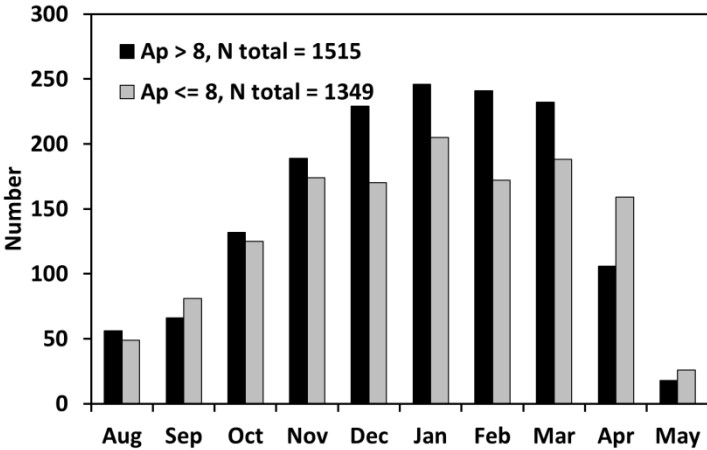

Figure 3: The number of measurements per month during the geomagnetic activity years (Ap > 8) – black columns, grey columns – the monthly distribution of measurements during geomagnetic quiet years (Ap <= 8).

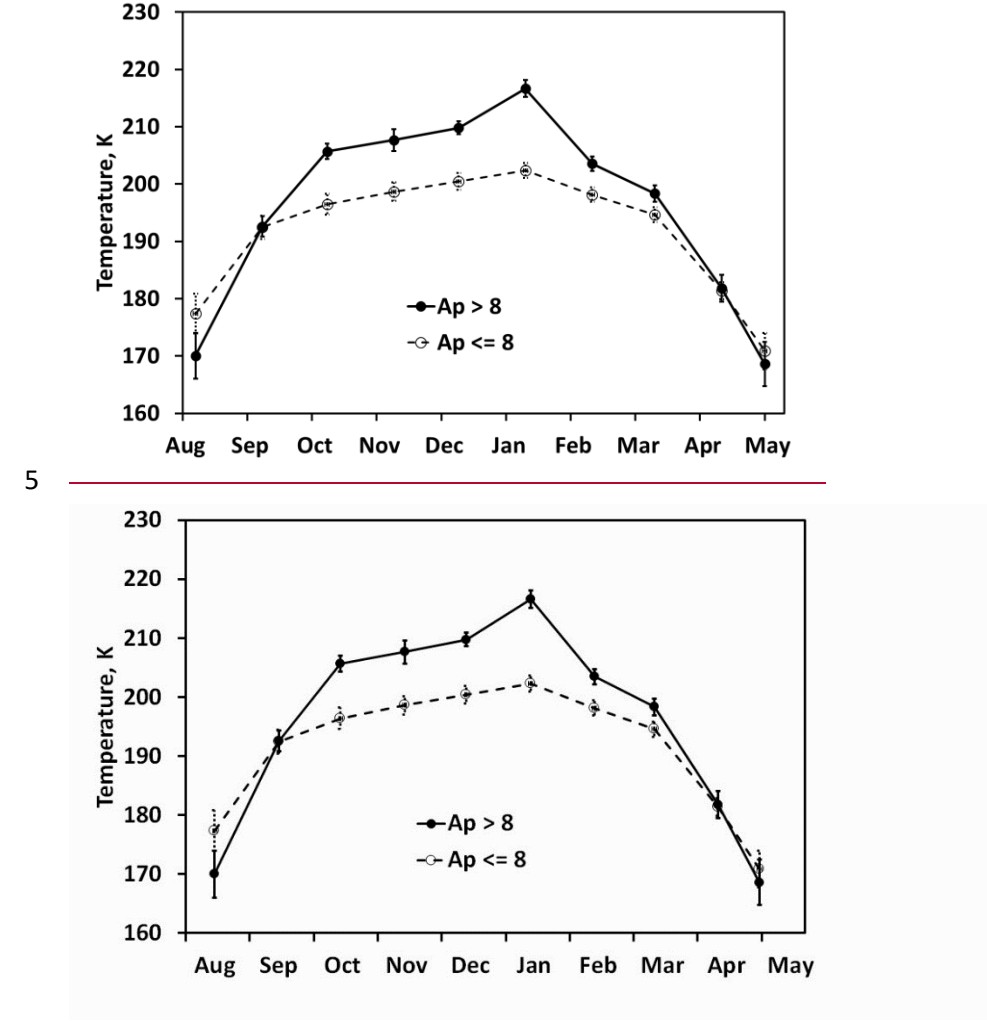

Figure 4: A monthly mean TOH in geomagnetic active years (Ap> 8) and in geomagnetic quiet years (Ap ≤ 8). Vertical bars correspond the standard deviations.

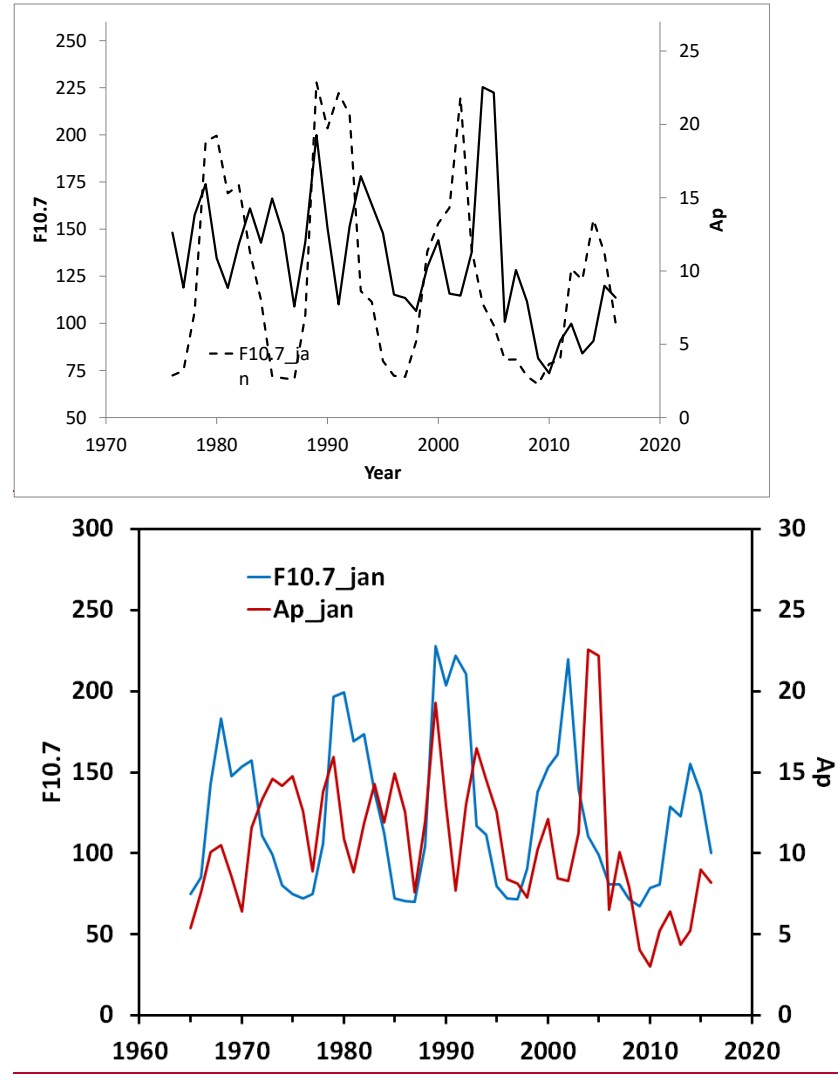

Figure 5: The F10.7 and Ap-index averages variations in January from 1975 to 2016.