# Peer review of "Influence of geomagnetic activity on mesopause temperature over Yakutia"

_Atmospheric Chemistry and Physics, 2017_

## Referee Comment (RC1) · Anonymous Referee #2 · 23 Oct 2017

The paper is based on a big data set of almost 2900 nights of data acquired over nearly 16 years. The results show higher monthly mean OH temperature with high Ap (>8) than with lower Ap (<=8) from October through January. The difference is about 10 K (i.e. 10.5K±1.4K, or 9.6K±1.4K, if Feb is included, according to what I have "measured" and calculated based on figure 4), but the text mentions 10 K only in the abstract and in the Conclusion (Page4, Line13), but not when figure 4 is explained. The difference approximately disappears in the remaining months of observation. This is the main finding.

[my "measurements" on figure 4 gave scale 50K/113mm, and oct: 20mm -> 8.9K, nov: 21mm -> 9.3K, dec: 21mm -> 9.3K, jan: 33mm -> 14.6K, feb: 13mm -> 5.8K]

I still find the treatment of the existing literature in the discussion section too long; it is

nearly a review, although by no means complete (compared to the additional literature cited in the recent and somewhat related paper by Yi et al., 2017). I think that at least part of this literature overwiew should go to the introduction, while skipping some of the details of how the literature results were obtained. At any rate, some improvement in structure (like subtitles for the different sub-topics, temperature effect from decrease of ozone radiative cooling - stratospheric warming - direct particle precipitation effects on temperature) would also be helpful.

[my Yi reference is Yi, W., Reid, I.M., Xue, X., Younger, J.P., Murphy, D.J., Chen, T., and Dou, X. (2017), Response of neutral mesospheric density to geomagnetic forcing, Geophys. Res. Lett. 44, 8647-8655, doi:10.1002/2017GL074813. While the focus is on density (with temperature only an auxiliary parameter), it cites many papers about solar activity effects via Joule and particle heating, and about geomagnetic forcing on ozone (none of which are mentioned in the present paper), stating that the expected temperature and density impact has "never been found".]

Only after 30 lines of "review", the topic returns to the authors' original results related to figures 4, and turns to the comparison of the long-term behaviour of F10.7 and the Ap index in figure 5. The rest of the discussion is again about literature results.

The long literature discussion is not as closely focussed on the observed 10 K winter enhancement for high Ap conditions as would be desirable, and some reorganization and editing would help for easier reading.

Minor details: Page1, Line32: add after "activity", ", the" -> "...measure of geomagnetic activity, the widely available Ap index..." ["index Ap" sounds as if its name were not well-known]. Since "index" is latin, the plural "indices" should be used (same line).

Fig. 1: the overlap between the F10.7 and Ap curves makes it not easy to read. Shifting the zero point for F10.7 upwards would help.

Page2, Line31: Missing "The" before "first group"; (same issue next sentence).

Line34: "approximately the similar" -> "approximately similar" (or "approximately the same"); change to read "geomagnetically active years" [an adverb, not an adjective].

Fig. 4: the tics on the time axis seem to be the beginning of each month; to make this easier to see, the labels should be centered between these ticks. On the other hand, the temporal positions of the Aug, Sep, and May data points look as if there was something wrong (not centered near mid-month).

Page3, Line32: This reference to the result given in the previous section (temperature enhancement due to geomagnetic activity) should be formulated so that it does not sound like news, here. Also, the emission height of OH has been mentioned before. Here, only the height difference of 7 km matters, so that the previous sentence could continue "...limited to a height of 80 km, which is 7 km below the hydroxyl emission layer".

Line34: "has to be detected"? The argument is that stratwarm effects are known to propagate downward, so that the OH temperature effect should be expected to occur earlier than model results obtained for 80 km, and below.

L35, 36: -> "measurements", delete "also", or start sentence with "Also, most of...".

L40: it would be better to connect both sentences with ", because in order to separate...", because they are related.

Page4, Line13: missing "is" between "mesopause" and "approximately", missing space between "10" and "K".

L16: "onset of warming was noticed", better "the average onset of stratospheric warmings is observed (Seppälä et al., 2013)" to be more explicit, and avoid the impression that the timing of stratwarms was unknown, before 2013.

L37: missing "i" in "Gavrilyeva".

The occurrences of "Seppälä" (the 2 in P3L22, and the ones in P3L27, P4L7, P4L16)

should be spelled correctly, as in P4L32, P5L10, P5L25, P5L27.

Page5, Line13: Mies title has "X dublett-Pi", and pages 150-188.

Line14,15: "III" is part of the family name, so -> "Russell III, J.M.". But, I ask myself (and the authors), wouldn't Randall, C.E., Harvey, V.L., Siskind, D.E., France, J., Bernath, P.F., Boone, C.D., and Walker, K.A.: NOx descent in the Arctic middle atmosphere in early 2009, Geophys. Res. Lett. 36, L18811,doi:10.1029/2009GL039706, 2009. be more pertinent than the Randall et al. 2007 paper, although it's "only" about special conditions in early 2009. (The "x" in NOx is subscript).

Line29: missing "." after "Kallenrode, M.-B".

%%%%%%%%%%%%%%%%%%%%%%%%%%%%%%%%%%%%%%%%%%%%%%%%%%%%%%%%%%%%%%%%

---

## Referee Comment (RC2) · Anonymous Referee #2 · 8 Nov 2017

In my opinion, the statement "The correlation coefficient is equal $0.51 \pm 0.1$ at 95% confidence level." (already in the abstract, Page1, Line16, and also Page2, Line30) is not meaningful. In both cases, the "confidence level" may be interpreted to refer to the meaning of the error bar, i.e., that 0.1 be a 2-sigma error bar, but it seems that the authors interpret more into their numbers than this. Namely, in the Conclusions (Page4, Line12), the authors say "Correlation .... is statistically significant and is equal to 0.51.", the error bar is not mentioned, and so the reader is expected to believe that the correlation coefficient of 0.51 itself "is meaningful".

However, as Aldrich (Aldrich, J. (1995), Correlations genuine and spurious in Pearson and Yule, Statistical Science 10(4), 364-376) explains,

"...there would be a correlation of about 0.4 to 0.5 between these indices had the bones

been distributed absolutely at random. (Pearson 1897).

The values of "about 0.4 to 0.5" came from a formula that Pearson developed for the correlation of x1/x3 and x2/x3 when x1, x2, and x3 are independent random variables with equal coefficients of variation."

[The Pearson paper mentioned is K. Pearson (1897), On a form of spurious correlation which may arise when indices are used in the measurements of organs, Proc. Roy. Soc. Lodon Ser. A, 60, 489-498, but according to the Proc.Roy.Soc. website, the correct year is 1896, not 1897]

In my opinion, this means that the level of correlation between Ap and OH temperature is well in the range of what statisticians have called "spurious", and by itself not a clear indication of a "real" effect. Only by geophysical arguments (as those which the authors do bring up) can the concept of a real relationship between geomag activity and mesopause region temperature be based.

―――――――――――――――――――――

---

## Referee Comment (RC3) · Anonymous Referee #1 · 22 Nov 2017

This paper is very interesting as it present observational evidence of different types of solar influence on middle/upper atmospheric temperatures. I have several recommendations for clarifying the language and would also suggest converting the figures to colour to make them easier to interpret before the paper can be accepted for publication. I have recommended major revisions, but this is mainly due to language improvements and the need to clarify certain aspects of the work before it will be clear enough to be published.

General comments:

Since much of the discussion focuses on the geomagnetic forcing and in particular energetic particle precipitation (EPP) impact on the mesosphere chemistry and the link to temperatures, it would be beneficial to have a short paragraph of the now well understood effects of EPP in the introduction going into the proposed temperature impacts via ozone modulation in the mesosphere. In the discussion, the link to temperatures is somewhat difficult to follow. I would recommend clarifying this following along the line of these steps:

1. EPP ionisation leads to production of both HOx and NOx species. This production can be proxied using indices such as Ap. (This is the main link to the Ap-temperature correlations of this study)

2. HOx and NOx contribute to ozone balance in the mesosphere and stratosphere. These effects are well known as demonstrated by the works sited in the existing text.

3. Model simulations have shown that the EPP driven ozone reduction in the polar winter upper mesosphere leads to reduction in long wave (terrestrial outgoing radiation) cooling. This signal is seen as increase of upper mesospheric temperature when comparing simulations with high EPP forcing to those with no, or low EPP forcing.

4. Higher Ap -> more EPP -> more HOx and NOx -> less ozone -> impact on polar winter mesospheric temperatures. This effect on temperatures is focused on polar winter atmosphere, which seems to be in a good agreement with the results presented in this manuscript.

It is not clear from the text presently how sensitive the layer of excited hydroxyl used for the temperature measurements is to changes in HOx concentrations i.e. those related to EPP. Could you please add a comment? This I think is needed to clarify to the readers weather the observed temperature changes are likely linked to changes in ozone of in HOx concentrations.

Specific comments and text revisions:

Page 1 L7: "beginning of the 24th" L11-13: "The maximum of the seasonally averaged temperatures is delayed by 2 years relative to the maximum of flux of radio emission from the Sun with a wavelength of 10.7 cm, and correlates with a change in geomagnetic activity. Ap-index as a measure of geomagnetic activity is taken." Change to "The maximum of the seasonally averaged temperatures is delayed by 2 years relative to the maximum of solar radio emission flux (wavelength of 10.7 cm), and correlates with a change in geomagnetic activity (Ap-index)."

L19-20: "The review of Beig et al. (2008) lists numerous studies showing that the response..."

L21: Add the abbreviation F10.7 here as it is used later: "solar radio flux at a wavelength of 10.7 cm in 10-22 W M-2 Hz-1 (F10.7)"

L22-23: "SABER radiometer onboard the TIMED satellite"

L23-24: "100 SFU, in agreement with the"

L26-30: You should make it clearer in this paragraph that the first studies only used a very short period of observations.

L35-36: I recommend revising this to: " As this is similar in scale to the observed delay of 25 months, it was logical to assume that the long-term temperature fluctuation of the subauroral mesopause correlates with the change in geomagnetic activity."

L37: "...between geomagnetic activity (Ap-index) and..."

Page 2 L2: "Mesopause (80-100 km) is the atmosphere region where the mesosphere borders on a thermosphere..."

L4: Does "activated" here refer to "exited"?

L4: "...hydroxyl molecule experiences..."

L8: "optical station Maimaga (63°N, 129.5°E) which is located at a distance of about 120 km to the north of Yakutsk" suggest changing to "optical station of Maimaga (63°N, 129.5°E) located about 120 km north of Yakutsk, Russia." Could you also give the magnetic latitude of the station?

L10: What is the significance of not having aurora present when the observations are made? This would have an impact on observing the direct EPP effect as particle precipitation can be associated with aurora displays.

L17-19: These 2 sentences are presently not clear.

L27: "…Ap-index mean values are shown…"

L30: "The correlation coefficient of TOH and Ap-index is equal $0.51 \pm 0.1$ at 95% confidence level." Remove word "equal". What is the correlation of F10.7 and TOH in your present dataset?

Paragraph starting at line 31: How were the two Ap groups selected, what is the transition value of 8 based on?

Page 3 L1: "…many papers have been published on the atmosphere response to solar and magnetospheric proton…"

L6-7: "Observations from satellites confirm that energetic particle precipitation changes the NOx amount in the atmosphere."

L8: "…from satellite measurements during the years 1992…"

L10: ECHAM5/MESSy is the same as the EMAC model, EMAC stands for "ECHAM5/MESSy Atmospheric Chemistry" i.e. 2 of the studies mentioned in the Discussion are from the same model.

L11-12: "They calculated thermospheric NOx fluxes to the mesosphere from precipitation of low-energy electrons using the average annual Ap from 1991 to 2005."

L12-14: "These average annual NOx concentrations were based on the UARS/HALOE measurements reported by Randall et al., (2007)." The NOx model of Baumgaertner et al. was based on the Randall et al. measurements, they were then compared with independent observations by the MIPAS instrument onboard Envisat as reported by Funke et al. (2005) (see reference in Baumgaertner et al., 2009).

L15-16: I think what you should say is that that the authors of that paper demonstrated that Ap works as a good proxy for low-energy produced NOx. That particular paper does not handle HOx at all. But there are others which show the direct impact of electron precipitation on HOx, for example: Andersson, M. E., P. T. Verronen, S. Wang, C. J. Rodger, M. A. Clilverd, and B. R. Carson (2012), Precipitating radiation belt electrons and enhancements of mesospheric hydroxyl during 2004–2009, J. Geophys. Res., 117, D09304, doi:10.1029/2011JD017246.

L17: They demonstrated both mesospheric and stratospheric ozone changes.

L21: "There is a publication series..." change to "There are several publications..."

L27-30: A similar downwards descending signal (in the same model) is already demonstrated by Baumgaertner et al (2011) using Geopotential height anomalies.

L30: "moves"

L32-33: Not all models are limited to this altitude range, but many reanalysis datasets are limited to altitudes below the stratopause. Models have issues in comprehensive inclusion of EPP.

L34: "Therefore, warming in our measurements has to be detected earlier." I don't understand why this would have to be the case. These temperature signals can be completely independent. That doesn't mean they would not be linked to geomagnetic activity or EPP.

Page 3 L16-17: "Warming signal moves down from high altitude to low one." and most of the last paragraph. This is not necessarily the case and certainly this is not a conclusion you can make based on the present study. Several of the publications you have cited actually argue that the stratosphere temperature signals are driven by changes in dynamics and are not related to in situ changes in ozone.

Figure 1: It is very difficult to tell the two lines apart, I would suggest making the plot in colour or applying an offset to separate the lines.

Additional typos and general language comments - "Energetic particle precipitation" (and the abbreviation EPP) is the generally used term. It is singular, therefore "EPP is. . .". Be careful not to use "energetic particles precipitation" or "particles precipitation", or "precipitations".

- NOx and HOx - the x is a subscript

- The commonly used terms for both are NOx = "Odd Nitrogen", HOx = "Odd hydrogen" instead of "nitrogen oxides" etc.

- The author with two papers in the citation list is "Seppälä", the name is correct in the citation list but incorrect in the text.

---

## Author Comment (AC1) · 28 Nov 2017

We are grateful for the thorough analysis of the manuscript and for the helpful and constructive comments.

The reviewer comments are given in normal typeface, ***our responses are italicized and bold.***

Responses:

Major finding: "… The difference is about 10 K (i.e. 10.5K±1.4K, or 9.6K±1.4K, if Feb is included, according to what I have "measured" and calculated based on figure 4), but the text mentions 10 K only in the abstract and in the Conclusion (Page4, Line13), but not when figure 4 is explained…".

***Thank you for your "measurement". We include it in figure 4 explanation.***

I still find the treatment of the existing literature in the discussion section too long; it is nearly a review, although by no means complete (compared to the additional literature cited in the recent and somewhat related paper by Yi et al., 2017). I think that at least part of this literature overview should go to the introduction, while skipping some of the details of how the literature results were obtained. At any rate, some improvement in structure (like subtitles for the different sub-topics, temperature effect from decrease of ozone radiative cooling - stratospheric warming - direct particle precipitation effects on temperature) would also be helpful. While the focus is on density (with temperature only an auxiliary parameter), it cites many papers about solar activity effects via Joule and particle heating, and about geomagnetic forcing on ozone (none of which are mentioned in the present paper), stating that the expected temperature and density impact has "never been found"

**Response: As we understood, the main shortcoming of the article is its structure. In the revised article, we tried to re-write the introduction and discussion according your remarks. Part of the discussion concerning an indirect effect on the atmosphere from particle penetration was significantly reduced and moved to the introduction. In the introduction and discussion, the main attention is paid to the response of the temperature of the upper mesosphere / lower thermosphere to geomagnetic activity. The references describing the direct effect of geomagnetic activity on the temperature of the upper atmosphere have been added to the introduction [Burns et al., 2014, Xu et al., 2013, Chang et al., (2009), and Jiang et al., (2014)].**

Minor details:

Page1, Line32: add after "activity", ", the" -> "...measure of geomagnetic activity, the widely available Ap index..." ["index Ap" sounds as if its name were not well-known]. Since "index" is latin, the plural "indices" should be used (same line). ***The proposed changes are made.***

Fig. 1: the overlap between the F10.7 and Ap curves makes it not easy to read. Shifting the zero point for F10.7 upwards would help. ***The figure is changed to the colour plot.***

Page2, Line31: Missing "The" before "first group"; (same issue next sentence). ***Corrections are made.***

Line34: "approximately the similar" -> "approximately similar" (or "approximately the same"); change to read "geomagnetically active years" [an adverb, not an adjective].
***Corrections are made.***

Fig. 4: the tics on the time axis seem to be the beginning of each month; to make this easier to see, the labels should be centered between these ticks. On the other hand, the temporal positions of the Aug, Sep, and May data points look as if there was something wrong (not centered near mid-month). ***The figure 4 is corrected.***

Page3, Line32: This reference to the result given in the previous section (temperature enhancement due to

geomagnetic activity) should be formulated so that it does not sound like news, here. Also, the emission height of OH has been mentioned before. Here, only the height difference of 7 km matters, so that the previous sentence could continue "...limited to a height of 80 km, which is 7 km below the hydroxyl emission layer". *The sentences are rewritten.*

Line34: "has to be detected"? The argument is that stratwarm effects are known to propagate downward, so that the OH temperature effect should be expected to occur earlier than model results obtained for 80 km, and below. *"has to be detected" is changed to your proposed sentence "stratwarm effects are known to propagate downward, so that the OH temperature effect should be expected to occur earlier than model results obtained for 80 km, and below".*

L35, 36: -> "measurements", delete "also", or start sentence with "Also, most of...". *We changed to "Also, most of …".*

L40: it would be better to connect both sentences with ", because in order to separate...", because they are related. *Sentence changed to: "The data of several solar cycles is necessary because to separate correctly the influence of these components."*

Page4, Line13: missing "is" between "mesopause" and "approximately", missing space between "10" and "K". *Corrected.*

L16: "onset of warming was noticed", better "the average onset of stratospheric warmings is observed (Seppälä et al., 2013)" to be more explicit, and avoid the impression that the timing of stratwarms was unknown, before 2013. *"onset of warming was noticed" replaced by "the average onset of stratospheric warmings is observed".*

L37: missing "i" in "Gavrilyeva". *Corrected.*

The occurrences of "Seppälä" (the 2 in P3L22, and the ones in P3L27, P4L7, P4L16) should be spelled correctly, as in P4L32, P5L10, P5L25, P5L27. *Corrected.*

Page5, Line13: Mies title has "X dublett-Pi", and pages 150-188. *Corrected.*

Line14,15: "III" is part of the family name, so -> "Russell III, J.M.". But, I ask myself (and the authors), wouldn't Randall, C.E., Harvey, V.L., Siskind, D.E., France, J., Bernath, P.F., Boone, C.D., and Walker, K.A.: NOx descent in the Arctic middle atmosphere in early 2009, Geophys. Res. Lett. 36, L18811, doi:10.1029/2009GL039706, 2009. Be more pertinent than the Randall et al. 2007 paper, although it's "only" about special conditions in early 2009. (The "x" in NOx is subscript*). "The Randall et al. 2007" paper is more pertinent because it was chronologically the first publication.*

Line29: missing "." after "Kallenrode, M.-B". *Corrected.*

---

## Author Comment (AC2) · 28 Nov 2017

The reviewer comments are given in normal typeface, *our responses are italicized and bold*.

In my opinion, the statement "The correlation coefficient is equal $0.51 \pm 0.1$ at 95% confidence level." (already in the abstract, Page1, Line16, and also Page2, Line30) is not meaningful. In both cases, the "confidence level" may be interpreted to refer to the meaning of the error bar, i.e., that 0.1 be a 2-sigma error bar, but it seems that the authors interpret more into their numbers than this. Namely, in the Conclusions (Page4, Line12), the authors say "Correlation .... is statistically significant and is equal to 0.51.", the error bar is not mentioned, and so the reader is expected to believe that the correlation coefficient of 0.51 itself "is meaningful".

However, as Aldrich (Aldrich, J. (1995), Correlations genuine and spurious in Pearson and Yule, Statistical Science 10(4), 364-376) explains, "...there would be a correlation of about 0.4 to 0.5 between these indices had the bones been distributed absolutely at random. (Pearson 1897).

The values of "about 0.4 to 0.5" came from a formula that Pearson developed for the correlation of x1/x3 and x2/x3 when x1, x2, and x3 are independent random variables with equal coefficients of variation."

[The Pearson paper mentioned is K. Pearson (1897), On a form of spurious correlation which may arise when indices are used in the measurements of organs, Proc. Roy. Soc. Lodon Ser. A, 60, 489-498, but according to the Proc.Roy.Soc. website, thecorrect year is 1896, not 1897]

In my opinion, this means that the level of correlation between Ap and OH temperature is well in the range of what statisticians have called "spurious", and by itself not a clear indication of a "real" effect. Only by geophysical arguments (as those which the authors do bring up) can the concept of a real relationship between geomag activity and mesopause region temperature be based.

Response:

***Thank you for noticing the inaccuracy in the description of the correlation and pointing to it. The correlation coefficient of TOH and Ap-index is 0.51. The significance of correlation coefficient was tested with 14 degrees of freedom T-test. The critical value of correlation coefficient is 0.46 at the 0.05 level of significance. TOH is not correlate with F10.7, because correlation coefficient 0.36 is less than critical value. The correlation coefficient increases to 0.65 when F10.7 leads the temperature by 2 years. We agree with you that "… correlation by itself not a clear indication of a "real" effect." Therefore, we attempted to present in the article the results of a study of the effect of geomagnetic activity on the temperature of the upper mesosphere made by other researchers.***

---

## Author Comment (AC3) · 28 Nov 2017

We thank you for your positive reception of the manuscript, and for the helpful and constructive comments. As we understood, major points raised by referees is manuscript structure. In the revised version, we tried to follow referee's recommendations.

The reviewer comments are given in normal typeface, ***our responses are italicized and bold.***

General comments:

Since much of the discussion focuses on the geomagnetic forcing and in particular energetic particle precipitation (EPP) impact on the mesosphere chemistry and the link to temperatures, it would be beneficial to have a short paragraph of the now well under stood effects of EPP in the introduction going into the proposed temperature impacts via ozone modulation in the mesosphere. In the discussion, the link to temperatures is somewhat difficult to follow. I would recommend clarifying this following along the line of these steps:

1. EPP ionisation leads to production of both HOx and NOx species. This production can be proxied using indices such as Ap. (This is the main link to the Ap-temperature correlations of this study)

2. HOx and NOx contribute to ozone balance in the mesosphere and stratosphere. These effects are well known as demonstrated by the works sited in the existing text.

3. Model simulations have shown that the EPP driven ozone reduction in the polar winter upper mesosphere leads to reduction in long wave (terrestrial outgoing radiation) cooling. This signal is seen as increase of upper mesospheric temperature when comparing simulations with high EPP forcing to those with no, or low EPP forcing.

4. Higher Ap -> more EPP -> more HOx and NOx -> less ozone -> impact on polar winter mesospheric temperatures. This effect on temperatures is focused on polar winter atmosphere, which seems to be in a good agreement with the results presented in this manuscript.

It is not clear from the text presently how sensitive the layer of excited hydroxyl used for the temperature measurements is to changes in HOx concentrations i.e. those related to EPP. Could you please add a comment? This I think is needed to clarify to the readers weather the observed temperature changes are likely linked to changes in ozone of in HOx concentrations.

***Response: In the revised article, part of the discussion concerning an indirect effect on the atmosphere from particle penetration was significantly reduced and moved to the introduction. In the introduction and discussion, the main attention is paid to the response of the temperature of the upper mesosphere / lower thermosphere to geomagnetic activity. This particular paper does not process HOx at all. We used the rotational temperature of OH (6-2) band as a proxy of the neutral temperature of the atmosphere. The auroral atomic oxygen line is superimposed on the OH spectrum so that it is impossible to correctly calculate the rotational temperature. Thus, the spectra obtained during the aurora were excluded from the study.***

Specific comments and text revisions:

Page 1 L7: "beginning of the 24th" L11-13: "The maximum of the seasonally averaged temperatures is delayed by 2 years relative to the maximum of flux of radio emission from the Sun with a wavelength of 10.7 cm, and correlates with a change in geomagnetic activity. Ap-index as a measure of geomagnetic activity is taken." Change to "The maximum of the seasonally averaged temperatures is delayed by 2 years relative to the maximum of solar radio emission flux (wavelength of 10.7 cm), and correlates with a change in geomagnetic activity (Ap-index)." **The sentence was corrected**.

L19-20: "The review of Beig et al. (2008) lists numerous studies showing that the response: : : ". **Corrected.**

L21: Add the abbreviation F10.7 here as it is used later: "solar radio flux at a wavelength of 10.7 cm in 10-22 W M-2 Hz-1 (F10.7)". **Abbreviation F10.7 was added.**

L22-23: "SABER radiometer onboard the TIMED satellite". **Corrected.**

L23-24: "100 SFU, in agreement with the". **Corrected.**

L26-30: You should make it clearer in this paragraph that the first studies only used a very short period of observations. **The sentence "This study only used a very short period of observations which coincided with the maximum of solar activity." is included.**

L35-36: I recommend revising this to: " As this is similar in scale to the observed delay of 25 months, it was logical to assume that the long-term temperature fluctuation of the subauroral mesopause correlates with the change in geomagnetic activity." **Sentence is changed according your recommendation.**

L37: ": : :between geomagnetic activity (Ap-index) and: : :" **The purpose of paper is described as: "to find geomagnetic signatures in night measurements of OH rotational temperature obtained for the period August 1999 to May 2015".**

Page 2 L2: "Mesopause (80-100 km) is the atmosphere region where the mesosphere borders on a thermosphere: : :" **Sentence is changed according your recommendation.**

L4: Does "activated" here refer to "exited"? *"***activated" is replaced by "excited".**

L4: ": : :hydroxyl molecule experiences: : :" ***"commits" is replaced by "experiences".***

L8: "optical station Maimaga (63◦N, 129.5◦E) which is located at a distance of about 120 km to the north of Yakutsk" suggest changing to "optical station of Maimaga (63◦N, 129.5◦E) located about 120 km north of Yakutsk, Russia." Could you also give the magnetic latitude of the station? ***The sentence is changed. Geomagnetic coordinates of Maimaga station are added in text.***

L10: What is the significance of not having aurora present when the observations are made? This would have an impact on observing the direct EPP effect as particle precipitation can be associated with aurora displays. ***The reason is described in the "Instrumentation and measurement technique" as: "The atomic oxygen line which arises at high auroral activity superimposes on OH(6-2) spectrum. To avoid systematic errors in evaluating the temperature because of this, the data obtained in the absence of aurora were selected for the analysis".***

L17-19: These 2 sentences are presently not clear. **These sentences are replaced by:" The temperature corresponding to that model spectrum, which deviates least from the real spectrum, by not more than the registration noise, is considered as a best fit to the real hydroxyl rotational temperature. The random errors in measuring the temperature are typically 2-10 K, depending on signal-to-noise ratio."**

L27: ": : :Ap-index mean values are shown: : :" **"… variations ..." is excluded.**

L30: "The correlation coefficient of TOH and Ap-index is equal 0.51 ± 0.1 at 95% confidence level." Remove word "equal". What is the correlation of F10.7 and TOH in your present dataset? ***The sentence is changed to text: "The correlation coefficient of TOH and Ap-index is 0.51. The significance of correlation coefficient was tested with 14 degrees of freedom T-test. The critical value of correlation coefficient is 0.46 at the 0.05 level of significance. TOH is not significantly correlate with F10.7, because correlation coefficient 0.36 is less than critical value. The correlation coefficient increases to 0.65 when F10.7 leads the temperature by 2 years.".***

Paragraph starting at line 31: How were the two Ap groups selected, what is the transition value of 8 based on? **"The average AP in the observation interval of about 8 was chosen as the transition value".**

Page 3 L1: ": : :many papers have been published on the atmosphere response to solar and magnetospheric proton: : :" *The phrase "In the last decade, many papers have been published on the atmosphere response to the proton and electron fluxes with various energies" is shifted to "Introduction". As was mentioned above, part of the discussion concerning an indirect effect on the atmosphere from particle penetration was significantly reduced and transferred to the Introduction. P3 L1-L5 were moved to Introduction. P3 L6-L20 were deleted.*

L6-7: "Observations from satellites confirm that energetic particle precipitation changes the NOx amount in the atmosphere." - *deleted*

L8: ": : :from satellite measurements during the years 1992: : :" - *deleted*

L10: ECHAM5/MESSy is the same as the EMAC model, EMAC stands for - *deleted*

"ECHAM5/MESSy Atmospheric Chemistry" i.e. 2 of the studies mentioned in the Discussion are from the same model. - *deleted*

L11-12: "They calculated thermospheric NOx fluxes to the mesosphere from precipitation of low-energy electrons using the average annual Ap from 1991 to 2005." - *deleted*

L12-14: "These average annual NOx concentrations were based on the UARS/HALOE measurements reported by Randall et al., (2007)." The NOx model of Baumgaertner et al. was based on the Randall et al. measurements, they were then compared with independent observations by the MIPAS instrument onboard Envisat as reported by Funke et al. (2005) (see reference in Baumgaertner et al., 2009). – *deleted*

L15-16: I think what you should say is that that the authors of that paper demonstrated that Ap works as a good proxy for low-energy produced NOx. That particular paper does not handle HOx at all. But there are others which show the direct impact of electron precipitation on HOx, for example: Andersson, M. E., P. T. Verronen, S. Wang, C. J. Rodger, M. A. Clilverd, and B. R. Carson (2012), Precipitating radiation belt electrons and enhancements of mesospheric hydroxyl during 2004–2009, J. Geophys. Res., 117, D09304, doi:10.1029/2011JD017246. *Yes, this particular paper does not process HOx at all. We used the rotation temperature OH (6-2) as a proxy of the neutral temperature of the atmosphere. The auroral atomic oxygen line is superimposed on the OH spectrum so that it is impossible to correctly calculate the rotational temperature. Thus, the spectra obtained during the aurora were excluded from the study.*

L17: They demonstrated both mesospheric and stratospheric ozone changes. – *deleted*.

L21: "There is a publication series: : :" change to "There are several publications: : :" – *corrected.*

L27-30: A similar downwards descending signal (in the same model) is already demonstrated by Baumgaertner et al (2011) using Geopotential height anomalies. *This sentence is included in the text of Discussion.*

L30: "moves" - *corrected*

L32-33: Not all models are limited to this altitude range, but many reanalysis datasets are limited to altitudes below the stratopause. Models have issues in comprehensive inclusion of EPP. *This sentence is rewritten as: "It should be noted that model and experimental researches of meteorological parameters are limited to a height below ~80 km."*

L34: "Therefore, warming in our measurements has to be detected earlier." I don't understand why this would have to be the case. These temperature signals can be completely independent. That doesn't mean they would not be linked to geomagnetic activity or EPP. *The phrase "Nevertheless, it cannot be ruled out that the temperature rise of the upper mesosphere in geomagnetic active years is due to the in situ effect of EPP" is incorporate to the Conclusion.*

Page 3 L16-17: "Warming signal moves down from high altitude to low one." And most of the last paragraph. This is not necessarily the case and certainly this is not a conclusion you can make based on

the present study. Several of the publications you have cited actually argue that the stratosphere temperature signals are driven by changes in dynamics and are not related to in situ changes in ozone.

Figure 1: It is very difficult to tell the two lines apart, I would suggest making the plot in colour or applying an offset to separate the lines. Additional typos and general language comments - "Energetic particle precipitation" (and the abbreviation EPP) is the generally used term. It is singular, therefore "EPP is: : :". Be careful not to use "energetic particles precipitation" or "particles precipitation", or "precipitations". ***Figure 1 and Figure 4 are colored. Abbreviation EPP is used.***

- NOx and HOx - the x is a subscript – ***corrected***.

- The commonly used terms for both are NOx = "Odd Nitrogen", HOx = "Odd hydrogen" instead of "nitrogen oxides" etc. – ***corrected***.

- The author with two papers in the citation list is "Seppälä", the name is correct in the citation list but incorrect in the text. –***corrected***.

---

## Author Response (AR2)

**Dear Referees,**

We are grateful for the thorough analysis of the manuscript, interested review which will be useful for our future studies.

Response to Referee#1:

 "Page 3 L16-17: "Warming signal moves down from high altitude to low one." and most of the last paragraph. This is not necessarily the case and certainly this is not a conclusion you can make based on the present study. Several of the publications you have cited actually argue that the stratosphere temperature signals are driven by changes in dynamics and are not related to in situ changes in ozone."

After some reflections, we agree with you that we cannot make a conclusion "Warming signal moves down from high altitude to low one" based on the present study. The layer of excited hydroxyl is rather narrow (3-7 km) and located in the upper mesosphere, while the downward movement of heat is found in the stratosphere. Therefore, we tried to remove all phrases concerning "Warming signal moves down..." (P4:L8-L11, P4:L15-L21, P4:L40-L41, P5:L1-L2 in the uncorrected version). Also, we do not refer to Karami et al. paper.

"If the amount of OH changes in the mesopause region, is the TOH affected? I think what you are telling me is "No", but I'm still not quite sure. This point would be worth saying clearly. We do not know for sure, but we think that there is a relationship:
 When the internal gravity wave (IGW) passes through the layer of excited hydroxyl, there is a connection between the intensity of radiation and the temperature introduced by Krasovskii: ΔI/I=ηΔT/T. The η parameter, which is called the Krassovskii number, determines the relationship between the relative intensity and temperature variations during adiabatic processes arising in the emission layer when an IGW propagates in it. A similar relation is probably true for waves of any scale. (Khomich et.al.: Airglow as an Indicator of Upper Atmospheric Structure and Dynamics, Springer-Verlag, Berlin, 740 pp., 2008). The auroral heating also can lead to an adiabatic process and change the amount of the excited hydroxyl.

In recent article Teiser, and von Savigny (J. Atmos. Solar-Ter. Phys., 2017, V. 161, p. 28-42. DOI: 10.1016/j.jastp.2017.04.010), studied the emission rate and altitude of the OH based on spaceborne nightglow measurements with the SCIAMACHY/Envisat. The SCIAMACHY observations cover the time period from August 2002 to April 2012. The analysis focused on low latitudes. They found some evidence for a 11-year solar cycle signature in the emission rate and in the emission altitude. As was mentioned in paper, a solar cycle signature in mesopause temperature was found by many researchers.

3. Talking about the two possible ways geomagnetic activity can modify temperatures in the mesopause region is very useful. What is not clear from the new text is in what timescales these effects work in? If strong geomagnetic events are excluded, does the joule heating effect persist beyond the events? Can we estimate the contributions of the different mechanisms for temperature changes to the TOH observations you report? A sentence or two in the conclusions would be very worth adding.

These questions are difficult to answer. Perhaps we can answer these questions later. We will study proton and energetic electron precipitation events signatures in our data using satellite data.

Response to Referee#2:

The paper has been successfully revised, removing my previous reservations, and is now fully acceptable for publication in ACP. There are only some very minor copy editing details left to repair (as far as I can tell from the "acp-2017-541-manuscript-version5.pdf" file which signals the changes with respect to the previous version, and still has some inconsistencies from leftovers), plus a few text improvements (for clarity) that I suggest in the following list.

The reviewer comments are given on the left column. Our responses are in the right column (pages and line numbers refer to the last corrected manuscript).

| Reviewer comment                                     | Responses                                            |
|------------------------------------------------------|------------------------------------------------------|
| P1: L9: -> "at the altitude of"                      | P1: L9: corrected                                    |
| L15: is the dot before "10 K higher" real?           | P1:L14: corrected                                    |
| L16: that should read "at the 95% level"             | P1:L16: corrected                                    |
| L20: is "of changes" before "in solar activity"      | P1:L16: "of changes" deleted                         |
| really necessary? If solar activity were constant,   |                                                      |
| we would not call it a cycle                         |                                                      |
| L26: could be shortened to read "to the change in    | P1:L24: corrected                                    |
| F10.7 flux", because it is clearly defined above     |                                                      |
| L29: better, change to "That study only used" to     | P1:L27: changed                                      |
| remove any doubt that the 2002 paper is referred     |                                                      |
| to, not the present manuscript.                      |                                                      |
| L35: better, change order to read "The changes of    | P1:L32-L35: Corrected "In this study as a measure    |
| F10.7 radio flux and the Ap index of magnetic        | of geomagnetic activity, the widely available        |
| disturbance over the last", to make clear that       | index Ap was used. The changes of F10.7 radio        |
| both parameters change (if emphasis on the           | flux and the Ap index of magnetic disturbance        |
| change is really necessary). However, the Ap         | over the last 4 cycles of solar activity is shown in |
| index is explained in the following sentence, so     | Figure 1."                                           |
| the order of presentation should be changed          |                                                      |
| P2:L3: correct "upper latitude atmosphere"           | P2:L3-L4: corrected                                  |
| (deleting "latitude"?), and "through the of          |                                                      |
| energetic"                                           |                                                      |
| L5: As individual papers are mentioned only some     | P2:L5-L6: corrected                                  |
| lines later, "In these papers" is not helpful, here. |                                                      |
| Better, concatenate with previous sentence "         |                                                      |
| energies, discussing the influence of geomagnetic    |                                                      |
| activity on atmospheric temperature in two           |                                                      |
| different ways".                                     |                                                      |
| L7: "The first way [or, maybe, "process"] is a       | P2:L8: replaced by "One process is"                  |
| direct effect".                                      |                                                      |
| L9: avoid "evidences", since it's used as not        | P2:L10: corrected                                    |
| countable -> "There is some evidence"                |                                                      |
| L10: put instrument/satellite in same order          | P2:L11: corrected                                    |
| "SABER/TIMED and MIPAS/Envisat":                     |                                                      |
| L13: -> "The authors suggested"                      | P2:L14: corrected                                    |
| L14: At start of sentence> "Von Savigny" or          | P2:L18-L21:                                          |
| rearrange so the reference appears later in the      | A significant decrease in the occurrence rate of     |
| sentence.                                            | noctilucent clouds in the southern polar             |
| 115: no need of plural in "the noctilucent cloud     | mesopause region was observed immediately            |
| occurrence rate", or change to read " in the         | after the onset of the enhanced solar particle       |
| occurrence rate of noctilucent clouds"; "made        | precipitation in SCIAMACHY (an imaging               |

| with SCIAMACHY" is misplaced here, and               | spectrometer installed on satellite Envisat) data  |
|------------------------------------------------------|----------------------------------------------------|
| sentence needs reordering (author X observed         | on January 16, 2005 by von Savigny et al., (2007). |
| something in satellite data after onset of EPP (?)   |                                                    |
| event, or something was observed after onset of      |                                                    |
| EPP in satellite data by author X). And, the next    | Yes                                                |
| sentence refers to the same finding (?).             |                                                    |
| L21: there should also be "by" before "Jiang et      | P2:L25: corrected                                  |
| al.", because they are two different and             |                                                    |
| independent papers.                                  |                                                    |
| L22: "low thermosphere temperature" ->               | P2:L26: corrected                                  |
| "temperature of the lower [!] thermosphere", to      |                                                    |
| avoid the misunderstanding of the temperature        |                                                    |
| being low.                                           |                                                    |
| L23: is this the "second way" (or, process)? Then,   | P2:L27: ", whose energy is deposited mainly " is   |
| you should say so (or replace "the first way" by     | changed to ". The energy of precipitating          |
| "one way [or process]. What does "whose              | particles is deposited mainly"                     |
| energy" refer to? "the energy of precipitating       |                                                    |
| particles that is deposited mainly"                  |                                                    |
| L31 (also L33): there should be no article before    | P2:L37: corrected                                  |
| "hydroxyl", or even reorder -> "hydroxyl             | P2:L39: corrected                                  |
| rotational temperature"                              |                                                    |
| L32: replace "height of its radiation" by "mean      | P2:L38: replaced by "mean emission height"         |
| emission height", "emission centroid height", or     |                                                    |
| "at about 87 km".                                    |                                                    |
| L36: avoid repetition of "widely available index".   | P2:L41: "As a measure of geomagnetic activity,     |
|                                                      | the widely available index Ap was used." -         |
|                                                      | deleted                                            |
| L38: "The mesopause", strictly speaking, is not a    | P3:L1: changed                                     |
| region but the precise altitude where the vertical   |                                                    |
| temperature gradient is zero (the height of which    |                                                    |
| varies). Change to refer to the mesopause region     |                                                    |
| (80 - 100 km).                                       |                                                    |
| L39: Avoid repetition; see above, L32.               | P3:L5: "at the radiation height" - deleted         |
| L42: I think that "see, e.g., " is needed before the | P3:L5: corrected                                   |
| Khomich reference to avoid misleading readers        |                                                    |
| about since when the OH temperature is used to       |                                                    |
| diagnose mesopause region temperature. And,          |                                                    |
| avoid misleading term "radiation height".            |                                                    |
| P3:L1: better, change "the presented paper" to       | P3:L6-L7: corrected                                |
| "this paper", or "the present paper". And, better    |                                                    |
| add "described by Ammosov and Gavrilyeva             |                                                    |
| (2000), so that the definite article before          |                                                    |
| "infrared spectrograph" makes sence (or else,        |                                                    |
| change to read "with an infrared spectrograph        |                                                    |
| (Ammosov)").                                         |                                                    |
| L5: -> "on the OH(6-2) spectrum                      | P3:L10: corrected                                  |
| L24: -> "The mean values of", but averaged           | P3:L26: Replaced by "The same TOH and Ap,          |
| over what? Aha, the same averages as in 2a!          | averaged over the same years, are shown in         |
| Better, reformulate to make this clear, and that     | Figure 2a."                                        |
| the same TOH data are repeated in 2a and 2b.         |                                                    |
| P4:L33: -> "et al."; "geopotential" (lower case!)    | P4:L12: corrected                                  |

| L36: repetion of emission height "The hydroxyl radiating (~ 87 km)" has been removed in new | These lines are deleted.                        |
|---------------------------------------------------------------------------------------------|-------------------------------------------------|
| text, L37 etc. Delete repeated lines of text.                                               |                                                 |
| P5:L9: "data" is plural, so -> "are necessary to                                            | P4:L31-L32: corrected                           |
| separate" (also, delete "because")                                                          |                                                 |
| L22: -> "in search for a geomag"                                                            | P5:L35: corrected                               |
| L27: -> "explained by the altitude difference                                               | The sentence is deleted                         |
| between the mesopause"                                                                      |                                                 |
| L29: -> "The warming signal"; hmmm, and "to                                                 | The sentence is deleted                         |
| low one" is not informative ("moves down" says                                              |                                                 |
| it all).                                                                                    |                                                 |
| several refs: missing hyphen in "J.Atmos.Solar-                                             | corrected                                       |
| Terr.Phys."                                                                                 |                                                 |
| L24: missing hyphens in "9-day" and "13.5-day"                                              | P5:L34: corrected                               |
| Caption Figure 3: "-black columns, grey columns-                                            | Changed to:                                     |
| ": confusing change of order (after/before main                                             | Figure 3: The number of measurements per        |
| explanation). If at all necessary (the symbols are                                          | month during the geomagnetic activity years (Ap |
| explained in the figure, clearly enough), "black                                            | > 8) and quiet years (Ap <= 8).                 |
| columns", "grey columns" should be included in                                              |                                                 |
| the parenteses "(Ap>8; black columns)", "(Ap<=8;                                            |                                                 |
| grey columns)".                                                                             |                                                 |
| Caption Figure 4: -> "Monthly mean TOH"                                                     | Corrected                                       |
| Caption Figure 5: -> "F10.7 and Ap-index averages                                           | Corrected                                       |
| for January", because the fact that they vary is                                            |                                                 |
| obvious from the figure.                                                                    |                                                 |

[revised manuscript text omitted]
 rise of the upper mesosphere in geomagnetically active years is due to the in situ effect of EPP.

**Acknowledgments.**

Russian Foundation for Basic Research supported the reported study according to the research projects No. 17-05-00855 A, 5 15-05-05320 A.

---

## Author Response (AR3)

Dear Referees,

We are grateful for the thorough analysis of the manuscript, interested review which will be useful for our future studies.
* * *
Response to Referee#1:

1. "Page 3 L16-17: "Warming signal moves down from high altitude to low one." and most of the last paragraph. This is not necessarily the case and certainly this is not a conclusion you can make based on the present study. Several of the publications you have cited actually argue that the stratosphere temperature signals are driven by changes in dynamics and are not related to in situ changes in ozone."
**After some reflections, we agree with you that we cannot make a conclusion "Warming signal moves down from high altitude to low one" based on the present study. The layer of excited hydroxyl is rather narrow (3-7 km) and located in the upper mesosphere, while the downward movement of heat is found in the stratosphere. Therefore, we tried to remove all phrases concerning "Warming signal moves down…" (P4:L8-L11, P4:L15-L21, P4:L40-L41, P5:L1-L2 in the uncorrected version). Also, we do not refer to Karami et al. paper.**

2. "If the amount of OH changes in the mesopause region, is the TOH affected? I think what you are telling me is "No", but I'm still not quite sure. This point would be worth saying clearly.
**We do not know for sure, but we think that there is a relationship:**
**When the internal gravity wave (IGW) passes through the layer of excited hydroxyl, there is a connection between the intensity of radiation and the temperature introduced by Krasovskii: ΔI/I=ηΔT/T. The η parameter, which is called the Krassovskii number, determines the relationship between the relative intensity and temperature variations during adiabatic processes arising in the emission layer when an IGW propagates in it. A similar relation is probably true for waves of any scale. (Khomich et.al.: Airglow as an Indicator of Upper Atmospheric Structure and Dynamics, Springer-Verlag, Berlin, 740 pp., 2008).**
**The auroral heating also can lead to an adiabatic process and change the amount of the excited hydroxyl.**
**In recent article Teiser, and von Savigny (J. Atmos. Solar-Ter. Phys., 2017, V. 161, p. 28-42. DOI: 10.1016/j.jastp.2017.04.010), studied the emission rate and altitude of the OH based on spaceborne nightglow measurements with the SCIAMACHY/Envisat. The SCIAMACHY observations cover the time period from August 2002 to April 2012. The analysis focused on low latitudes. They found some evidence for a 11-year solar cycle signature in the emission rate and in the emission altitude. As was mentioned in paper, a solar cycle signature in mesopause temperature was found by many researchers.**

3. Talking about the two possible ways geomagnetic activity can modify temperatures in the mesopause region is very useful. What is not clear from the new text is in what timescales these effects work in? If strong geomagnetic events are excluded, does the joule heating effect persist beyond the events? Can we estimate the contributions of the different mechanisms for temperature changes to the TOH observations you report? A sentence or two in the conclusions would be very worth adding.
**These questions are difficult to answer. Perhaps we can answer these questions later. We will study proton and energetic electron precipitation events signatures in our data using satellite data.**

Response to Referee#2:

The paper has been successfully revised, removing my previous reservations, and is now fully acceptable for publication in ACP. There are only some very minor copy editing details left to repair (as far as I can tell from the "acp-2017-541-manuscript-version5.pdf" file which signals the changes with respect to the previous version, and still has some inconsistencies from leftovers), plus a few text improvements (for clarity) that I suggest in the following list.

The reviewer comments are given on the left column. Our responses are in the right column (pages and line numbers refer to the last corrected manuscript).

| Reviewer comment | Responses |
| --- | --- |
| P1: L9: -> "at the altitude of" | P1: L9: corrected |
| L15: is the dot before "10 K higher" real? | P1:L14: corrected |
| L16: that should read "at the 95% level" | P1:L16: corrected |
| L20: is "of changes" before "in solar activity" really necessary? If solar activity were constant, we would not call it a cycle... | P1:L16: "of changes" deleted |
| L26: could be shortened to read "to the change in F10.7 flux", because it is clearly defined above | P1:L24: corrected |
| L29: better, change to "That study only used..." to remove any doubt that the 2002 paper is referred to, not the present manuscript. | P1:L27: changed |
| L35: better, change order to read "The changes of F10.7 radio flux and the Ap index of magnetic disturbance over the last...", to make clear that both parameters change (if emphasis on the change is really necessary). However, the Ap index is explained in the following sentence, so the order of presentation should be changed | P1:L32-L35: Corrected "In this study as a measure of geomagnetic activity, the widely available index Ap was used. The changes of F10.7 radio flux and the Ap index of magnetic disturbance over the last 4 cycles of solar activity is shown in Figure 1." |
| P2:L3: correct "upper latitude atmosphere" (deleting "latitude"?), and "through the of energetic..." | P2:L3-L4: corrected |
| L5: As individual papers are mentioned only some lines later, "In these papers" is not helpful, here. Better, concatenate with previous sentence "... energies, discussing the influence of geomagnetic activity on atmospheric temperature in two different ways". | P2:L5-L6: corrected |
| L7: "The first way [or, maybe, "process"] is a direct effect...". | P2:L8: replaced by "One process is..." |
| L9: avoid "evidences", since it's used as not countable -> "There is some evidence..." | P2:L10: corrected |
| L10: put instrument/satellite in same order "SABER/TIMED and MIPAS/Envisat"; | P2:L11: corrected |
| L13: -> "The authors suggested..." | P2:L14: corrected |
| L14: At start of sentence, -> "Von Savigny...", or rearrange so the reference appears later in the sentence.
L15: no need of plural in "the noctilucent cloud occurrence rate", or change to read "... in the occurrence rate of noctilucent clouds"; "made | P2:L18-L21:
A significant decrease in the occurrence rate of noctilucent clouds in the southern polar mesopause region was observed immediately after the onset of the enhanced solar particle precipitation in SCIAMACHY (an imaging |

| | |
|---|---|
| with SCIAMACHY" is misplaced here, and sentence needs reordering (author X observed something in satellite data after onset of EPP (?) event, or something was observed after onset of EPP in satellite data by author X). And, the next sentence refers to the same finding (?). | spectrometer installed on satellite Envisat) data on January 16, 2005 by von Savigny et al., (2007).

Yes |
| L21: there should also be "by" before "Jiang et al.", because they are two different and independent papers. | P2:L25: corrected |
| L22: "low thermosphere temperature" -> "temperature of the lower [!] thermosphere", to avoid the misunderstanding of the temperature being low. | P2:L26: corrected |
| L23: is this the "second way" (or, process)? Then, you should say so (or replace "the first way" by "one way [or process]. What does "whose energy" refer to? "the energy of precipitating particles that is deposited mainly..." | P2:L27: ", whose energy is deposited mainly… " is changed to ". The energy of precipitating particles is deposited mainly..." |
| L31 (also L33): there should be no article before "hydroxyl", or even reorder -> "hydroxyl rotational temperature" | P2:L37: corrected
P2:L39: corrected |
| L32: replace "height of its radiation" by "mean emission height", "emission centroid height", or "at about 87 km". | P2:L38: replaced by "mean emission height" |
| L36: avoid repetition of "widely available index". | P2:L41: "As a measure of geomagnetic activity, the widely available index Ap was used." - deleted |
| L38: "The mesopause", strictly speaking, is not a region but the precise altitude where the vertical temperature gradient is zero (the height of which varies). Change to refer to the mesopause region (80 - 100 km). | P3:L1: changed |
| L39: Avoid repetition; see above, L32. | P3:L5: "at the radiation height" - deleted |
| L42: I think that "see, e.g., " is needed before the Khomich reference to avoid misleading readers about since when the OH temperature is used to diagnose mesopause region temperature. And, avoid misleading term "radiation height". | P3:L5: corrected |
| P3:L1: better, change "the presented paper" to "this paper", or "the present paper". And, better add "described by Ammosov and Gavrilyeva (2000), so that the definite article before "infrared spectrograph" makes sence (or else, change to read "with an infrared spectrograph (Ammosov...)"). | P3:L6-L7: corrected |
| L5: -> "on the OH(6-2) spectrum | P3:L10: corrected |
| L24: -> "The mean values of...", but averaged over what? Aha, the same averages as in 2a! Better, reformulate to make this clear, and that the same TOH data are repeated in 2a and 2b. | P3:L26: Replaced by "The same TOH and Ap, averaged over the same years, are shown in Figure 2a." |
| P4:L33: -> "et al."; "geopotential" (lower case!) | P4:L12: corrected |

| | |
|---|---|
| L36: repetion of emission height "The hydroxyl radiating ... (~ 87 km)" has been removed in new text, L37 etc. Delete repeated lines of text. | These lines are deleted. |
| P5:L9: "data" is plural, so -> "are necessary to separate..." (also, delete "because") | P4:L31-L32: corrected |
| L22: -> "in search for a geomag..." | P5:L35: corrected |
| L27: -> "explained by the altitude difference between the mesopause..." | The sentence is deleted |
| L29: -> "The warming signal..."; hmmm, and "to low one" is not informative ("moves down" says it all). | The sentence is deleted |
| several refs: missing hyphen in "J.Atmos.Solar-Terr.Phys." | corrected |
| L24: missing hyphens in "9-day" and "13.5-day" | P5:L34: corrected |
| Caption Figure 3: "-black columns, grey columns-": confusing change of order (after/before main explanation). If at all necessary (the symbols are explained in the figure, clearly enough), "black columns", "grey columns" should be included in the parenteses "(Ap>8; black columns)", "(Ap<=8; grey columns)". | Changed to:
Figure 3: The number of measurements per month during the geomagnetic activity years (Ap > 8) and quiet years (Ap <= 8). |
| Caption Figure 4: -> "Monthly mean TOH..." | Corrected |
| Caption Figure 5: -> "F10.7 and Ap-index averages for January...", because the fact that they vary is obvious from the figure. | Corrected |

[revised manuscript text omitted]